# Radiation and Radical Grafting Compatibilization of Polymers for Improved Bituminous Binders—A Review

**DOI:** 10.3390/ma17071642

**Published:** 2024-04-03

**Authors:** Wiktoria Baranowska, Magdalena Rzepna, Przemysław Ostrowski, Hanna Lewandowska

**Affiliations:** 1Centre for Radiation Research and Technology, Institute of Nuclear Chemistry and Technology, 16 Dorodna St., 03-195 Warsaw, Poland; 2ORLEN Asfalt sp. z o.o., 39 Łukasiewicz St., 09-400 Płock, Poland; 3Department of Transportation Engineering, Faculty of Civil and Environmental Engineering, Gdansk University of Technology, 80-233 Gdansk, Poland; 4School of Health & Medical Sciences, University of Economics and Human Sciences in Warsaw, Okopowa 59, 01-043 Warsaw, Poland

**Keywords:** bituminous binders, asphalt mixture, polymer modification, road pavement deterioration, additives and modifiers, colloidal dispersion system, grafting

## Abstract

This review scrutinizes current research on new methods for enhancing bituminous binder performance through radiation and radical grafting of polymer modifiers of bitumen. It investigates innovative methods, including using waste polymers as modifiers and applying radiation for polymer grafting, to overcome challenges like high costs, low aging resistance, and storage stability issues, of which separation of phases polymer/bitumen is the most significant obstacle. These advanced modification techniques promise sustainability through the decrease of the carbon footprint of transportation systems by improving the properties and durability of binders. Additionally, this review discusses the parameters and mechanistic aspects from a scientific perspective, shedding light on the underlying processes that contribute to the improved performance of modified bituminous binders.

## 1. Introduction

Bitumens of various types and grades are primarily applied in road construction, as well as railways, hydrotechnics, isolation materials, and chemical industries [1,2]. In road construction, bitumen plays a crucial role as a binder in the asphalt mixtures that are used to pave road surfaces. Although bitumen constitutes a relatively small mass proportion of the entire asphalt mixture (4–6% (*w*/*w*)), its impact on the durability of the road surface is significant [3,4].

In recent years, the increasing transport and communication needs of the global economy and society are increasing pressure to expand, upgrade, and maintain the world’s road network to a high level of safety and standard of travel. As a consequence of increasing transport and environmental needs, the durability of pavement structures is becoming increasingly important, which translates into road network reliability [5]. As a crucial and integral part of the transportation system, road surfaces play a significant and measurable role in promoting socio-economic sustainability [6]. It is proven [7,8] that improving the performance of bituminous binders reduces the need for frequent road maintenance and repair and can lead to a decrease of the carbon footprint of the transport system, aligning with principles of sustainability.

In terms of chemistry, the bituminous binder is an extremely complex mixture of high-molecular-weight hydrocarbons that contain heteroatoms such as nitrogen, oxygen, and sulfur, along with trace amounts of metals like vanadium and nickel [2,9]. The complexity of the chemical nature of bitumens makes them extremely difficult to understand [3], and they have yet to be fully discovered and characterized [6]. Several scientific theories have been proposed to describe the structure of crude oil and its refining products, including bitumen [10,11,12,13,14]; among them, the most widespread is the theory that describes bitumens as a colloidal dispersion system in which asphaltenes, stabilized by resins, are dispersed in an oil medium consisting of saturates and aromatics [9]. The relative proportions of these components determine the phase stability of bitumen, the loss of which affects the deterioration of bituminous binders’ properties.

In order to improve the performance properties of asphalt mixtures and extend the service life of asphalt surfaces, various types of additives and modifiers are used for bituminous binders [2,15,16]. The most commonly used modifiers and additives to improve the properties of bitumens are polymers, chemical modifiers, fibers, fillers, antioxidants, natural bitumens, adhesion improvers, viscosity modifiers, and recently increasingly popular recycled materials [17] and environmentally friendly renewable natural materials [18]. Polymer modification is a frequently used approach to improve the performance of bitumen. The use of polymer-modified binders (PMBs) in asphalt mixtures offers numerous benefits, such as their lower thermal susceptibility and greater resistance to rutting, low-temperature cracking, and fatigue [19], which results in the reduction of life cycle costs of asphalt pavements [20]. Elastomers, plastomers, and recycled tire rubbers are the most frequently introduced polymers. The viscosity modifiers and reactive polymers are being used less commonly [2]. Although polymers improve bitumen properties to some extent, there are still some drawbacks limiting the future development of polymer-modified bitumens (PMBs), including high cost, low aging resistance of polymers, and difficulty with storage stability [17]. Recently, researchers have investigated the use of waste polymers as modifiers to reduce expenses and enhance environmental protection [21,22,23,24,25]. However, various modifiers pose different challenges associated with their integration into bitumen. Modified bitumens, which contain various additives, exhibit a tendency to lose stability when stored at high temperatures for an extended time. This phase separation issue significantly limits their practical applications [26,27].

The crucial parameter is chemical compatibility between modifiers and bitumen. Issues arise most commonly due to the two-phase nature of polymer-rich and asphaltene-rich mixtures, leading to thermodynamic instability and potential phase separation [28]. Achieving compatibility involves different approaches, including cross-linking with bitumens molecules, surface activation treatment, additives-grafting, or hybrid approaches involving more than one of the aforementioned methods [26,27,29]. There are many reviews devoted to the modification of bitumens with various types of virgin and waste polymer additives and their influence on bituminous binder performance [21,22,23,24,25,30,31,32,33]. The authors of this review focused on a range of modifications frequently involving radical and radiation-induced processes that offer viable ways for addressing the challenge of bitumen–polymer system compatibility. Examples of these novel and developing approaches will be reviewed herein.

## 2. Electromagnetic and Electron Beam Radiation

Gamma rays, X-rays, UV radiation, and non-zero resting mass electrons are forms of ionizing radiation employed in industrial polymer modification. Ionizing radiation refers to radiation that has enough energy to remove electrons from atoms, leading to the formation of ions. In the context of industrial polymer modification, gamma rays and electron beams are particularly utilized. While photons are forms of electromagnetic radiation, manifesting as quantum mechanical excitations of the electromagnetic field and behaving as particles with zero mass, zero charge, and spin one, the electrons are elementary particles with a negative electric charge. Electrons possess rest mass, and their behavior is governed by both classical and quantum mechanics. Electrons engage with matter through scattering and ionization, and they can be redirected by electric and magnetic fields. In contrast, photons interact with matter through the photoelectric effect, Compton scattering, and pair production [34]. Unlike electrons, they lack an electric charge and remain unaffected by electric or magnetic fields. This results in different characteristics of dose deposition [35]. Electrons, even in the highest-utilized energies in industry, below 10 MeV (to minimize the unwanted here Bremmsstrahlung) exhibit limited penetration depth (nm-cm depending on energy) as they rapidly lose energy through interactions with matter, unlike photons, which therefore penetrate in a wider range depending on energy (from between 20 and 150 μm for UV up to several meters for gamma) [36].

Nevertheless, despite these differences, all types of ionizing radiation induce a similar chemical effect by dislodging electrons with energies significantly higher than the energy associated with an electron in a molecular or atomic orbital (the ionization energy of an electron in an atom typically ranges between 4 and 20 eV depending on the atom, while ionizing radiation can deliver energies in the keV or MeV range [37]). The dislodging of electrons by ionizing radiation leads to the generation of secondary electrons, which can generate ion pairs and free radicals, further leading to chemical bond breakage and formation. Besides the production of single-atom or low-molecular-mass products, ionizing radiation generates macroradicals located on the polymer chain, a recombination of which can result in chain branching, cross-linking, or scission. The prevalence of each reaction is influenced by multiple factors, including reactive species concentration and reaction kinetics [38]. These are derived from the kinetics of individual reactions, the content and permeability of gases (oxygen) in polymer mixtures, temperature of treatment, and dose rate, as well as the energy and type of radiation used [39].

## 3. Radiation Treatment and Grafting for Tailored Bitumen Modifiers

The utilization of high-power electron accelerators in industrial irradiation processes is appealing due to their significantly high throughput rates, and the treatment costs per unit of product frequently rival those of conventional chemical processes [35,38]. Radiation treatment is applied mostly to the bitumen modifiers, enhancing surface properties through processes like oxidation, degradation, and grafting.

### Radiation-Induced Grafting (RIG)

One approach to enhancing the performance of modified bitumen that is currently gaining attention is through the use of radiation and radical grafting of polymers, which can significantly improve their properties over a very wide range. In the following subsections, the findings have been described, to highlight the potential of radiation and radical grafting of polymers as a means of enhancing the performance of modified bituminous binders.

Grafting is a versatile technique in polymer science, which involves attaching one polymer (the “graft”) onto another polymer matrix (the “backbone”) [40]. This process allows for the modification and enhancement of the base polymer’s properties, introducing new characteristics and functionalities. Applications range from altering solubility, nano-dimensional morphology, and compatibility to improving charge transport properties. Various chemical approaches, such as free radical reactions, click reactions, amide formation, and alkylation, are employed for polymer grafting. Typically, these chemical methods are both expensive and complex, often requiring the use of various solvents and specific reactants. In contrast, radiation-induced grafting (RIG) presents a more practical and cost-effective approach, particularly when applied as a method for bitumen modification [39]. RIG is a technique known for its reduced reliance on additional chemical additives and solvents [40]. RIG can be accomplished using various ionizing radiations with different energies, including gamma, X-ray, electron beam (keV to MeV), and ultraviolet radiation (a few eV to about 100 eV) [39]. These forms of radiation are capable of penetrating various materials of petroleum origin to varying depths or interacting only with their surfaces, which can be effectively utilized for modifying these materials, offering precision and versatility in grafting [28,41,42]. Additionally, microwave radiation, known for its efficient and controlled heating properties, serves as another valuable tool in the arsenal of radiation-induced grafting methods [43,44,45]. Currently, microwave-induced devulcanization and pyrolysis are viewed as promising methods, particularly for the modification of waste tire rubber recyclate for bitumen additives [46]. The described exploration will center on how these methods can enhance polymer structures and introduce new functionalities, particularly within the realm of bituminous binder performance. Radiation-induced grafting, in particular, can modify surface properties without altering the bulk properties of the base polymer, making it advantageous for a wide range of industries. It is important to mention that various radiation-based initiation methods, including γ-irradiation and UV radiation, can be employed, offering versatility and expanded possibilities in polymer synthesis. These methods are also cost-effective because they operate at low temperatures and do not require catalysts or initiators. The combination of these techniques opens up new avenues for the controlled synthesis of graft copolymers in an industrially achievable manner, with a specific focus on improving bituminous binder performance.

## 4. Poly(styrene-butadiene-styrene) (SBS) and Its Modification

Modified bitumens, which contain various additives, are developed to improve their performance. Poly(styrene-butadiene-styrene) (SBS) block copolymer-modified bitumen is the binder of choice in asphalt mixtures for paving roads subjected to heavy traffic. This preference is attributed to its exceptional performance in road durability, providing superior resistance to the stresses of high-volume vehicular loads [30,47,48,49,50]. SBS, a triblock copolymer, exhibits a biphasic morphology where rigid polystyrene (PS) forms a dispersed phase within the flexible polybutadiene (PB) continuous phase [4]. At typical pavement operating temperatures, the PS blocks are in a glassy state, lending strength to SBS, while the PB blocks remain in a rubbery state, adding elasticity [51,52],. When SBS is compatible with bitumen, the PB blocks absorb maltenes, leading to swelling, whereas the PS blocks undergo minimal swelling and serve as physical cross-linking points [53]. This interaction results in a two-phase morphology between the swollen SBS and bitumen. With lower concentrations of SBS, it is dispersed within the continuous bitumen phase. However, as the concentration of SBS increases, a phase inversion (PI) occurs. This transition leads to the formation of two interlocked continuous phases, which is considered the ideal phase morphology for SBS-modified bitumen (Figure 1) [51]. Within the range of the glass transition temperatures (Tg) of PS blocks (approximately 100 °C) and PB blocks (around −90 °C), PS exhibits a glassy behavior while PB remains rubbery [54], facilitating the formation of physical cross-links primarily involving entanglements and interactions between the polystyrene (PS) end-blocks [55]. This capacity to form physical cross-links positions SBS as a prime contender for bitumen modification. SBS in the bitumen matrix improves its viscoelastic properties, making the bitumen more durable and resistant to temperature variations, rutting, and cracking. This network also contributes to better handling and application properties of the bitumen during road construction by improving adhesion and allowing longer workable time [47,49].

However, one major challenge with bitumen/SBS blends is their tendency to lose stability when stored at high temperatures for extended time. This tendency may depend on the chemical nature of the base bitumen and the properties of the polymers used. This phase separation issue significantly limits their practical applications [26,27]. Nonetheless, researchers have been increasingly focused on improving thermal stability during the storage of SBS-modified binders [8,27,56,57,58,59,60].

### 4.1. Radiation-Induced Grafting of SBS

Fu et al. [27] investigated the use of methacrylic acid as a polar monomer to modify SBS using ^60^Co γ-rays radiation (SBS-g-M). The grafting degree achieved was approximately 6% for SBS-g-M. The impact of SBS-g-M on the mechanical properties of the binder was evaluated using a dynamic shear rheometer (DSR), both before and after grafting. The results showed that the addition of SBS-g-M had a significant effect on the rheological properties of the binder, leading to improved high-temperature performance and reduced temperature susceptibility compared to SBS-modified bitumen. Specifically, the G/sinδ parameter was used as a criterion for assessing the properties of the binder at high temperature. The temperature at which G/sinδ was equal to 1000 Pa was 67.5 °C for base bitumen, 78.2 °C for polymer modified bitumen (PMB) with the addition of 5.0% SBS, and 82.4 °C for PMB with the addition of 5.0% SBS-g-M, respectively (Figure 2).

It is important to note that the rheology of modified bitumen is dependent on the polymer content. When the polymer content is sufficient (>6%), the phase angle (δ) decreases, and the complex modulus (G*) increases considerably. Higher SBS-g-M content results in a stable tan δ but increasing complex modulus due to polymer network formation. Direct observation of the morphology using a fluorescence microscope revealed that the compatibility between SBS-g-M and bitumen was effective, and the storage stability of the binder was significantly improved compared to standard SBS-modified bitumen. Furthermore, the use of SBS-g-M led to smaller particle sizes and more homogeneous dispersion in the bitumen matrix than non-grafted SBS, under the same high temperatures and high shear stress. The addition of SBS-g-M to bituminous binder significantly improves its properties, as the blended system can form a more robust network in the modified bitumen. This network structure gives the binder superior high-temperature performance and reduced temperature susceptibility.

In order to improve the mechanical properties and high-temperatures resistance of PMB, various additives, including fibers, are also used. In a recent study [61], the researchers used UV radiation to induce a “thiol-ene click reaction” through the creation of the covalent bonds of vinyl-terminated dendritic polyester (VTDP) and polyester (PET) fibers to create a hyperbranch-structured PET fiber called VTDP-PET. The PET or VTDP-PET fiber was then blended with SBS-modified bitumen to make fiber/SBS-modified binders. The preparation of VTDP-PET fiber involved a two-step method. First, OH and SH functional groups were introduced to the surface of PET fiber through chemical modification. Secondly, a dendritic VTDP was grafted onto the modified PET fiber using a “thiol-ene click reaction” (Figure 3). Chemical analyses, including FTIR, XPS, and SEM, showed that the polymer-activating groups were successfully grafted onto the PET fibers, resulting in increased surface roughness and shape gradient.

The addition of VTDP-PET or PET fibers significantly improved flow resistance, viscoelasticity, and rutting resistance of PMB modified by SBS. Notably, VTDP-PET/SBS-modified binders exhibited more significant enhancement than PET/SBS binders. Additionally, VTDP-PET/SBS-modified bitumen showed higher thermal stability, as demonstrated by TGA, compared to a PET/SBS-modified binder. The shear stress, storage modulus (G′), and loss modulus (G″) at 46 °C of 2.5% VTDP-PET/SBSMA were 64%, 54.9%, and 76.9% higher than those of 2.5% PET/SBSMA, respectively. Overall, VTDP-PET fiber offers a promising alternative for creating modified bitumen for heavy-traffic roads.

### 4.2. Chemical Grafting of SBS

SBS grafting typically involves the use of chemical initiators, with radiation initiators being used less frequently. Overall, chemical modification of elastomers involves the use of various grafting monomers, most commonly including maleic anhydride (MAH), maleic acid (MA), dibutyl maleate (DBM), and acrylic acid (AA) and its esters [62,63]. Recent research has focused on carbon-based nanomaterials, such as carbon nanotubes (CNTs). CNTs have been shown to reduce fatigue and permanent deformation in hot mix asphalt (HMA) compared to conventional HMA [64]. CNTs can also improve thermal cracking resistance and reduce bitumen aging [65,66]. However, the high cost of CNTs makes them unsuitable for a large-scale use in asphalt pavements. This motivates the search for more cost-effective carbon-based nanomaterials for asphalt roads. One potential candidate is graphite nanoplatelets (GNPs) [66,67,68,69]. GNPs are nano-discs with a sub-micrometer diameter and a nanometer thickness. GNPs can be produced from either graphene or natural graphite. GNPs produced from graphene typically consist of several layers of graphene sheets, each of which is a single layer of carbon atoms. Depending on the type and carbon purity of GNPs, their cost is comparable to some existing bitumen modifiers such as SBS and significantly lower than the cost of multi-wall CNTs [67]. Han et al. [70] proposed a composite material consisting of polystyrene-grafted graphene nanoplatelets (PS-GNPs) that were prepared and used as a modifier for SBS in bitumen preparation. Polystyrene-grafted graphene nanoplatelets (PS-GNPs) were prepared from graphene oxide by ultrasonic shock-induced grafting of styrene, followed by reduction of the oxygen functional groups with hydrazine. For the research, the bitumen was modified using 5% SBS, which had been pre-modified with GNPs or PS-GNPs within the range of 0% to 0.04% admixture. The modification of SBS influenced its ability to increase bitumen ductility (by up to 50% for 0.05% GNP and 20% for PS-GNP-modified SBS) and softening point temperature (by ca. 8% for both modifications) relating to SBS alone. It also reduced penetration (by 15% and 9% for GNP- and PS-GNP-modified SBS, respectively). In summary, the polymer stabilization of graphite nanoplatelets appears to enhance hardness and thermal stability of the resulting modified bitumen. A closer examination of the micro-properties of the PS-grafted GNPs (Figure 4) provides valuable insight into their macroscopic behavior, offering understanding of how these microscopic changes translate to altered macro properties. Young’s modulus and reduced modulus of GNPs were 2180 MPa and 2395 MPa, respectively, and for PS-GNPs, were correspondingly reduced to 586 MPa and 644 MPa (372% decrease compared to GNPs). This suggests that the modified GNPs became more deformable, indicating an increase in toughness and a decrease in rigidity. The authors proposed that the change in properties is mainly attributed to the expanded layer spacing and the PS coating on the GNPs forming a spring-like structure. Additionally, the PS grafted onto the GNPs has similar styrene groups to those in SBS (Styrene-Butadiene-Styrene), facilitating easy physical entanglement with the PS phase in SBS macromolecules and thereby significantly enhancing dispersion in SBS-modified bitumen. The addition of PS-GNPs into bitumen has improved the bitumen binder capacity to resist both pressure and permanent deformation.

Another study by Han et al. [71] proposed a modifier of SBS waste rubber powder (WRP) with octadecyl-amine (ODA) through a covalent grafting reaction. The ODA-WRP was then combined with SBS to prepare ODA-WRP/SBS-modified bitumen. In the process, the amine group (–NH_2_) present in ODA reacted with the carboxyl group (–COOH) on the surface of WRP, leading to the formation of an amide (–NHCO–), which facilitated the attachment of ODA to the surface of WRP. The mechanical properties of the designed PMB showed significant improvement as compared to the pristine SBS-modified bitumen, as confirmed by various tests, including DSR, MSCR, and separation tests. The enhancement in properties was attributed to the better compatibility of various components in bitumen and the establishment of a network structure between ODA-WRP and SBS (Figure 5).

The successful synthesis of ODA-WRP was confirmed through FT-IR and SEM analyses. The study has proven potential applications of ODA-WRP binders in constructing highways and other infrastructure. The value of complex modulus G* for ODA-WRP (0.5:20)-modified bitumen was 5.81 kPa at 10 Hz, which is approximately 36.47% higher than that of WRP-modified bitumen. This indicates a considerable increase in the bitumen’s stiffness and its resistance to deformation, making it more resilient under load. Regarding the multiple creep and recovery (MSCR) test, at a lower stress of 0.1 kPa and a higher stress of 3.2 kPa (at 82 °C), the J_nr_ value (non-recoverable creep compliance) for ODA-WRP (0.5:20)-modified bitumen was the smallest among the tested samples at 0.153 kPa^−1^ and 1.328 kPa^−1^, respectively, which is about 59.20% and 37,48% lower than that of WRP-modified bitumen (J_nr0.1_ = 0.375 kPa^−1^ and J_nr3.2_ = 2.124 kPa^−1^). This substantial reduction in J_nr_ indicates an enhanced ability to resist permanent deformation under loading, making it highly suitable for applications in road construction where durability under heavy traffic is critical. Segregation tests indicated that the modified ODA-WRP exhibited enhanced compatibility in SBS-modified bitumen. However, it should be borne in mind that the achieved differences in softening temperatures for the upper and lower parts of samples are still at a high level, and such binders may not meet the requirements of European standards, as EN 14023 [72].

In another case [63], the SBS was modified with glycidyl methacrylate (GMA) using dicumyl peroxide (DCP) as an initiator at a temperature of 170 °C. The study aimed to investigate the impact of different factors such as % GMA, % DCP, and reaction time on grafting degree, conversion of GMA, and cross-linking using a factorial design approach. This provides a good opportunity to illustrate the influence of both concentrations and grafting factors on the grafting outcome and its impact on the material properties. Grafting increased with increasing GMA content. Yet, the concentration of GMA did not significantly affect the conversion and torque. The latter where highly influenced by the concentration of the initiator, DCP, within the studied range. DCP content had a strong positive effect on the final torque due to the cross-linking reactions of the matrix SBS. It was not possible to increase the grafting degree without increasing the torque level due to cross-linking reactions. This is an important consideration when designing the synthesis, as the initiator can significantly alter the properties of SBS in addition to grafting GMA. An optimum % DCP point of 0.1% *w*/*w* was identified, which allowed for the highest incorporation and conversion values at a low cross-linking level (Figure 6). Subsequent testing of the elaborated modified material in bitumen applications for durability and performance would indicate the usefulness of this method for road applications.

Li et al. [57] developed a novel bitumen modifier by reacting thiolated graphene oxide nanoplatelets (gOs-SH) and SBS using a thiol-ene click reaction, which is initiated by ammonium persulfate (APS). They tested the temperature resistance and mechanical properties of the PMB using a dynamic shear rheometer (DSR) and multiple stress creep recovery (MSCR) tests. Characterization of gOs-SH was conducted using various techniques such as FTIR, EDX, AFM, XRD, and SEM. The thiol-ene click reaction was elucidated using XPS. The study found that an optimal amount of gOs-SH was 0.02%, which improved the anti-rutting performance of the bitumen. Results confirmed that SBS-g-gOs exhibited excellent stability and dispersion in the base bitumen and that the addition of gOs-SH promoted the formation of an SBS network structure (Figure 7). Overall, the use of click chemistry for the preparation of gOs-modified-SBS bitumen was deemed a promising approach for industrial applications. The fluorescence microscopy results highlighted that the addition of GOs-SH promoted the formation of a more pronounced SBS network structure within the modified bitumen than unmodified SBS. The incorporation of GOs-SH into SBS-modified bitumen showed an improvement in ductility and softening point, with an optimal effect observed at 0.04% GOs-SH content. This modification resulted in an increase in softening point and ductility by about 9.14% and 12.35%, respectively, compared to the original SBS-modified bitumen. Conversely, penetration observed a decrease, followed by a continual increase, with 0.02% GOs-SH. The DSR tests revealed that the addition of 0.02% GOs-SH content resulted in the largest improvement in storage modulus (G′). At a temperature range of 46 °C to 82 °C, the 0.02% GOs-SH-modified bitumen exhibited an increase in G′, highlighting enhanced high-temperature stability compared to SBS and other GOs-SH concentrations. The MSCR test results demonstrated that the Jnr values (non-recoverable creep compliance) for 0.02% GOs-SH(SBS) were the lowest among the tested samples, showing reductions of about 35.38% and 39.06% under different stress levels compared to SBS and 0.04% GOs-SH(SBS)-modified bitumens. This indicates a considerable enhancement in the bitumen’s ability to resist permanent deformation, attributing to the added GOs-SH. In segregation experiments, the difference between the upper and lower softening points was reduced from 4.3 °C for the original SBS-modified bitumen to 2.5 °C for the 0.02% GOs-SH(SBS) modified bitumen, revealing that the addition of GOs-SH improved the stability and dispersion of modifiers in the bituminous matrix.

Wang et al. [73] synthesized SBS copolymers with different functional groups, including end amino, carboxylic acid, and hydroxyl, using 1,5-diazabicyclo[3.1.0]hexane, carbon dioxide, and epoxy ethane as capping agents, respectively. The study aimed to investigate the impact of these end-polar groups on the morphology and dynamic mechanical properties of the copolymers. The results of morphology showed that the PS domains changed from uniform spheres in SBS to disordered, incompact strips in end-functionalized SBS (Figure 8). Furthermore, the study concluded that end amino and carboxylic acid were effective in improving the storage stability and compatibility of SBS-modified binders, as evidenced by comparing the softening points of end-functionalized and non-functionalized SBS-modified bitumens. The SBS-modified bitumen with end amino (SBS–N-modified bitumen) and carboxylic acid (SBS–COOH-modified bitumen) groups showed remarkably improved stability, with a softening point difference between the top and bottom parts of the sample, of only 0.2 °C and 1.5 °C for SBS–COOH and N-COOH modified bitumens, respectively, versus 31.5 °C for the SBS-modified bitumen without functionalization.

Trials of grafting SBS with most commonly used monomers, such as GMA [63] or maleic anhydride (MAH) [74], in the melt using peroxides as initiators had low grafting efficiencies and could lead to undesired cross-linking.

## 5. Radiation Treatment and Grafting Strategies for Reclaimed Polymers

The use of polymer-modified binders (PMBs) in asphalt mixtures offers numerous benefits, such as their lower thermal susceptibility and greater resistance to rutting and fatigue. Nonetheless, their higher cost presents a limiting factor to their widespread use [21,48]. On the other hand, while polymers are known for their endurance and accessibility, a downside to their usage is the accumulation of substantial quantities of complex and challenging-to-handle waste with high molecular weights [22,75]. In recent times, researchers have investigated the possibility of utilizing waste polymers as modifiers in order to reduce expenses and promote environmental sustainability [22,23,75,76]. Consequently, the topic of using waste polymers for bituminous binders is currently the subject of many review papers [21,22,23,76].

The use of recycled polymers presents a cost-effective alternative to raw materials in PMB, which can extend the life of road asphalt pavements. It can be used in all categories of roads and layers. Unfortunately, a significant amount of plastic waste ends up in landfills without any recycling process [21]. However, due to the EU’s Green Deal climate policy, it can be seen that the amount of recycled polymers is increasing, e.g., in 2021, 5.5 million tons of post-consumer recycled plastics were re-introduced into the EU27+3 economy, an increase of 20% compared to 2020 (European production of plastics reached 57.2 Mt in 2021) [77]. Polymer waste processing methods vary depending on the origin, composition, and properties of the waste polymers [78]. There are different types of polymer waste processing. Although chemical conversion for reusing waste polymers is preferable, their high chemical resistance makes this approach challenging. The already-mentioned insufficient chemical bonding between the reclaimed polymer and bitumen, which may lead to separation of phases [28], is the most significant obstacle. As a result, radiation technologies are gaining popularity as they offer innovative ways to modify the properties of polymers, resulting in unique plastics and composites while adhering to green chemistry principles [22,75].

The reuse of waste polymer, especially the popular PP and PE recyclate materials in road pavements, is an issue of numerous reviews and current large-scale investigations, including the most recent ones [23,24,79,80,81,82,83,84,85,86,87,88,89]. The most commonly studied waste polymers as bitumen modifiers are high-density polyethylene (HDPE), low-density polyethylene (LDPE), polypropylene (PP), polystyrene (PS), polyethylene terephthalate (PET), Ethyl Vinyl Acetate (EVA), polyvinyl chloride (PVC), and ground tire rubber (GTR). Waste plastics can be effectively reused in high-value applications in highway construction, particularly in hot mix asphalt (HMA) [21,23]. The percentage of different polymers by weight of bitumen vary between 0.5 to 25%; they most commonly reach 2–7% for PE, PP, and EVA and 5–15% for GTR [21]. Polyolefins, such as PE and PP, are widely utilized thermoplastics and constitute a substantial part of recyclable plastic waste. They possess properties similar to rubber and fiber that can improve the characteristics of polymer-modified bitumen (PMB) used in asphalt pavement. Nevertheless, including PP and PE materials is a problematic issue, regarding their low compatibility with bitumen [86,88]. Due to their high crystallinity, they can cause issues with the bitumen matrix, resulting in phase separation, decreased cohesion, and the requirement for longer mixing times. Chemical modification of the polymer can enhance compatibility with bitumen [90]. Many chemical modifications have been proposed, especially for PP and PE [86], including grafting polyethylene with maleic anhydride (MA-g-PE) [86] and melt grafting of maleic anhydride onto low-density polyethylene/polypropylene blends [82]. Additionally, the discussion extends not only to various chemical modifications but also to the incorporation of other additives [86] and to innovative physical blending treatments [82].

The following subsections describe selected applications of grafting compatibilization of waste polymers to bituminous binders used to improve the properties of modified bituminous binders, with particular reference to radiation grafting.

### 5.1. Waste Polypropylene (PP)

Günay et al. [42] conducted a study to improve the compatibility of waste polypropylene with bitumen used gamma irradiation. The process involved modifying polypropylene by grafting maleic anhydride (MA) onto the polymer surface. Dicumyl peroxide was used as an initiator, and MA was dissolved in acetone and mixed into PP granules. The pre-treated PP granules were then exposed to gamma rays for 28 min with a dose of 10 kGy and monitored with a chemical dosimeter (Figure 9). Unreacted MA was removed by washing with water and drying. This process aimed to provide a chemical interaction between the modifier polymer and bitumen. Irradiated waste polypropylene (PP_R_-γ-MA) was then used as a bitumen modifier.

The physical properties of PP_R_-γ-MA-modified binders were tested with conventional methods. Various rheological methods were also used to determine low- and high-temperature performance, as well as fatigue resistance, using a LAS test. The MA grafting of waste PP improves its interaction with bitumen. As confirmed by FTIR spectroscopy findings, this process creates carbonyl groups on the grafted MA (or carboxyl groups in acid form) and an activated surface that contains radicals and oxygen-containing groups due to γ-irradiation in an air atmosphere. The authors suggest that the carbonyl groups of PP_R_-γ-MA will react with the asphaltenes in bitumen during the hot mixing process, which will facilitate the effective distribution of the polymer modification within the bitumen. Addition of PP-γ-MA to the bituminous binder results in a stiffer material with decreased penetration and ductility, increased softening point, and improved temperature susceptibility. However, the mixing and compaction temperatures calculated on the basis of the rotational viscosity of bitumen increase with higher levels of modification, suggesting a limitation on excessive use of PP-γ-MA. Gamma-irradiation reduces phase separation between the waste PP and bitumen, leading to better compatibility and physical performance of bitumen. However, using an excessive amount of waste polymer (7–9%) in bitumen can increase phase separation, regardless of gamma irradiation. Thus, it is recommended to use a maximum of 5% of PP_R_-γ-MA as a modifier in bitumen to achieve optimal results. The high-temperature performance of bituminous binder is improved by waste PP, with better elastic response compared to base bitumen and increased fatigue damage resistance. The MSCR test findings showed that the use of PP_R_-γ-MA results in decreased non-recoverable compliance J_nr_ and increased elastic recovery R of the material. The low-temperature performance grade (PG) of bitumen decreases after modification, but overall performance is improved. The findings suggest that waste PP can be used in bitumen with improved performance after MA grafting via gamma-irradiation. The results of the study indicated that using PP_R_-γ-MA as a modified material in binders is feasible and has the potential to improve fatigue life and resistance to damage.

As for an instance of chemical grafting modification, an interesting amphiphilic system was recently reported by Malus et al. [91]. Hydroxyl-functionalized isotactic polypropylene (FPP) was created using solution polymerization. This was then attached to an alternating styrene-co-maleic anhydride copolymer via a transesterification reaction, to yield FPP-g-SMA, amphiphilic graft-copolymers with varying contents of comonomers. Adding FPP and FPP-g-SMA30/35S modifiers significantly raised the softening point (the highest increase in softening point was achieved with the incorporation of 2.5 wt% FPP-g-SMA30S into bitumen, 62.0 °C to 76.9 °C) and improved the morphology of the end products. After the addition of FPP or FPP-g-SMA into bitumen, the heterogeneous “bee” microstructures, which are characteristic of unmodified bitumen, disappeared completely. The greatest compatibilizing impact was observed with FPP and FPP-g-SMA30S modifiers, which resulted in a fine, interlocked polymer distribution in the bitumen matrix, leading to improved rheological properties [91].

### 5.2. Waste Polyethylene (PE)

Ahmedaze et al. have carried out a number of studies on the application of radiation to enhance the properties of various types of polyethylene [28,41,92].

In one of their studies [28], electron beam irradiation (EBI) was applied to improve the properties of recycled high-density polyethylene (HDPE_R_) obtained from a waste bottle, which was used as a modifier for bitumen. The HDPE_R_ was irradiated with a dose of 20 kGy to improve chemical interaction and adhesion in the polymer/bitumen interface. Linear acceleration of electrons was used as a source of electrons with an energy of 4 MeV (information provided by the authors on request, for the purpose of this review). The results showed that the irradiated HDPE_R_ modification improved the compatibility of the components in the bitumen. The HDPE_Re_ polymer phases ranged in diameter from 3 to 20 μm and were well dispersed within the continuous bituminous phase. The polymer phase contained some dispersed bitumen inclusions, indicating a typical microphase separated structure. Notably, each particle in the bitumen-swollen polymer phase was surrounded by a dark brown narrow ring, signifying an interface formed by the chemical grafting of both components. This result aligned with FTIR spectroscopy, confirming the chemical interaction between HDPE_Re_ and bitumen components. Furthermore, DSC analysis revealed partial mixing and improved compatibility of the blend components. The DSR tests indicated that HDPE_Re_ modification significantly raises the complex modulus (G*) and reduces the phase angle (δ) values of the bitumen. The high-temperature performance grade (PG) of bitumen improves significantly after modification, increasing from PG 52-Y for the base bitumen to PG 82-Y with the highest degree of HDPR_Re_ modification of bitumen. The bitumen modified with 9% of electron beam-irradiated recycled high-density polyethylene showed the most significant increase in the softening point from 38.7 °C for the base bitumen to 90.1 °C for the modified one. In turn, the 7%-HDPR_RE_-modified bitumen in aged condition obtained the most significant reduction in δ values from 87.1 degrees for the base bitumen at 52 °C to 64.5 degrees at 82 °C, indicating an improvement in the elastic response. As a result, the HDPE_Re_-modified bitumen displays improved resistance to temperature-related changes and greater elasticity in response to shear stress compared to the base bitumen. HDPE_Re_ modification effectively reduces the temperature susceptibility of bitumen at high temperatures due to the stiffening effect of the HDPE_Re_ modifier, albeit with a slight adverse impact on creep stiffness and m-value, resulting in a slight decrease in the low-temperature PG of the binders, from PG X-28 for the base bitumen to PG X-16 for the 9%-HDPE_RE_-modified bitumen. Therefore, the use of irradiated HDPE_R_ modifiers in cold regions should be limited to avoid thermal cracking.

In another study [41], the aforementioned authors applied electron beam irradiation (EG-2.5 Van-de-Graf accelerator 2.5 MeV) with a dose of 20 kGy to modify recycled low-density polyethylene (e-LDPE_R_), which resulted in the formation of free radicals and certain functional groups that could potentially assist in the development of strong chemical bonds between the polymer modifier and bitumen (Figure 10). Particularly, in the FTIR spectrum of recycled LDPE (LDPE_R_), the authors observed the presence of a peak at 1147 cm^−1^. This feature, indicative of the tertiary carbon group (CH_3_)_2_C [93], is not typically found in the spectrum of pristine LDPE. The emergence of the CH_3_ group in LDPE_R_ is attributed to the degradation of polymer chains, prompted by UV irradiation from sunlight, which generates free radicals leading to alterations in the macromolecular structure, including some branching. Under the exposure of LDPE_R_ to electron beam irradiation, the occurrence of a band at 1157 cm^−1^ in e-LDPE_R_, along with the reduction of the band at 1147 cm^−1^, was attributed to the increase of unsaturation in the regions of tertiary carbon atoms (formation of (CH_3_)_2_-C=CH- group) [93]; hydrogen abstraction is more favorable at carbon sites with higher degrees of substitution [94]. Additionally, a novel band at 1086 cm^−1^ in the e-LDPE_R_ spectrum denotes the formation of secondary alcohol groups, likely resulting from the radiation-induced oxidation [93].

The authors suggested that the reactive chemical bonds and secondary hydroxyl groups formed during irradiation can bond with bitumen components during mixing. The DSC results indicated improved blend compatibility between bitumen and e-LDPE_R_ due to the formation of a mixed amorphous phase. Fluorescent microscopy confirmed the chemical interaction between bitumen and the polymer phase. Conventional tests showed improved stiffness and reduced aging effects on bitumen after modification with e-LDPE_R_. Based on fundamental rheological parameters, and high mixing and compacting temperatures, it is suggested that e-LDPE_R_ can be used as a modifier at a level of 5% polymer in bitumen for optimized performance in flexible pavements.

In another work [92], the researchers investigated the use of ion-irradiated recycled high-density polyethylene (i-HDPE_R_) as a modifier in bitumen, showing an increase in the rutting parameter with a gradual increment in i-HDPE_R_ modification. Tests of physical properties exhibit a decrease in penetration values from 195.5 [dmm] in base bitumen to 30 [dmm] with 9% i-HDPE_R_, indicating increased stiffness. The softening point rises from 38.7 °C to 72.1 °C, confirming the stiffening effect. DSR results further confirm these findings, presenting greater complex modulus (G*) and reduced phase angle (δ) across all frequencies for i-HDPE_R_ modified bitumens, indicating increased resistance to deformation and enhanced elasticity. These studies of Ahmedaze et al. [92] highlight the potential of using waste polymers in bitumen through the ion irradiation process; however, special attention should also be paid to the low-temperature properties, due to the significant stiffness of the binder.


*Chemically grafted LDPE compatibilization*


Another approach attempting a reduction in phase separation of LDPE was achieved with a combination of a classic additive, sulfur, and an emerging new radically grafted polymer blend compatibilizer PE-g-MA [95,96]. The enhancement in the softening point of LDPE-modified bitumen with the addition of compatibilizers was significant; for 3% LDPE-modified bitumen, the softening point increased from 57.2 °C to 65.1 °C when 1% PE-g-MA and 0.15% sulfur were added. For 6% LDPE, the softening point rose from 72.3 °C to 80.9 °C with the addition of the compatibilizers, and the most dramatic increase was observed for 12% LDPE, where the softening point went from 86.3 °C to a notable 120.5 °C with the addition of compatibilizers. Also, the difference in softening point values between the top and bottom portions of the test tube after a standard storage stability test were significantly decreased, from 15.9 °C for 12% LDPE without compatibilizers to 7.25 °C with 1% PE-g-MA and 0.15% sulfur. The LDPE-based blends and those with PE-g-MA and sulfur showed major weight loss at higher temperature ranges (380–520 °C for LDPE-only blends and 400–540 °C for blends with compatibilizers), reflecting improved thermal stability [96].

In another study [97], the PMB is combined with LDPE or GMA-g-LDPE and bitumen. GMA-g-LDPE-modified bitumen exhibits a higher softening point than LDPE-modified bitumen, resulting in improved rutting resistance at high temperatures. In comparison to the base bitumen, the softening points of LDPE-modified bitumen with varying content increase by 2.8 °C, 4.5 °C, 5.4 °C, and 10 °C, while those of GMA-g-LDPE-modified bitumen increase by 13.5 °C, 22.5 °C, 43.1 °C, and 49.7 °C, respectively. The ductility of GMA-g-LDPE-modified bitumen surpasses that of LDPE-modified bitumen. With increasing LDPE content, the ductility of the PMB decreases, whereas the behavior is opposite for GMA-g-LDPE-modified bitumen. Results indicate noticeable separation and aggregation of LDPE in LDPE-modified bitumen, revealing its instability at high temperatures. This instability occurs due to the automatic shrinkage movement of LDPE molecules, causing phase separation from the bitumen as temperatures rise. Conversely, GMA-g-LDPE-modified bitumen exhibits minimal temperature differences. With GMA-g-LDPE content ranging from 3% to 6% by weight, temperature differences of 0.8 °C, 1.1 °C, 1.5 °C, and 2.5 °C are observed, indicating stable performance at high temperatures.

### 5.3. Waste Rubber

Rubber constitutes a significant component of municipal solid waste and must be recovered after disposal [98]. The use of crumb rubber (CR) or ground tire rubber (GTR) from scrap tires as a modifier for paving bitumen has become popular due to its sustainability benefits, resulting in a product called rubberized bitumens. However, the compatibility issues between GTR and bitumen have limited its applications [99]. Cross-linking during vulcanization prevents the chemical degradation of GTR particles in bituminous binders, leading to poor cross-linking between GTR and bitumen [100]. This is due to the incompatible cross-linked network of GTR. To achieve the desired compatibility, GTR needs to be uniformly dispersed and bonded with the molecules of the bituminous binder, creating a homogeneous system [29]. To solve this problem, surface activation methods have been proposed to modify the chemical and physical characteristics of the GTR surface [99], including mechanical, microwave, low-temperature plasma (LTP), plasticizer, and other methods [29,46]. On the other hand, using high-energy electrons, X-rays, and UV or visible light could also serve to initiate the reaction. There are several ways to create initial points on GTR, such as exposing it to radiation, using radical initiators (either photochemical or thermal), acid etching, hydroxylation, and living radical initiation techniques like atom transfer radical polymerization initiators [29,101]. These processes have been successful in grafting monomers such as styrene, allylamine, acrylamide, and methacrylate onto the GTR surface [102]. Although grafting is more effective in achieving compatibility, it typically involves multiple steps of synthesis and purification procedures, making it challenging to apply in industry. Therefore, the combination of surface activation treatment of GTR and additives-grafting appears to be a potential approach for improving the compatibility of rubberized binder [29]. Moreover, mixing reclaimed rubber with new polymer materials reduces the overall production costs and helps minimize environmental pollution [98]. Many papers are available on various methods that could try to resolve the problems with the poor compatibility thermoplastic polymer/GTR systems [103,104,105,106,107], also by using radiation techniques–gamma radiation [98,108,109], including water-medium ionizing radiation treatment [110] and electron beam [111,112,113]. The following review focuses on the application of gamma and electron beam processing of waste tire rubber to produce rubberized binders.


**
*
Gamma and electron beam processing of ground tire rubber
*
**


Various radiation methods have been used to activate the surface ground tire rubber (GTR) particles, with or without air present, as stated above. The number of radicals produced on the surface of the rubber particles depends on several factors, such as the type of radiation used, the radiation dose, and the duration of the radiation exposure. The irradiation process typical for polymers follows four reaction steps: chain initiation, propagation, chain termination, and chain branching, according to the Bolland–Gee scheme (Figure 11). During the initiation step, hydrogen is abstracted from the GTR particle surface, resulting in the formation of two radicals (GTR· and H·). The GTR radical (the most likely carbon- and sulfur-centered mixture of distinct radicals [114]) can undergo various reactions, including reacting with oxygen to form peroxy radicals, propagation for polymer growth, chain termination, and chain branching (Figure 11) [102].

Gamma rays and polymers undergo an interaction that generates free radicals, which can result in chain branching, cross-linking, or scission. The prevalence of each reaction is influenced by multiple factors, including reactive species concentration and reaction kinetics [101]. In a study conducted by Ibrahim et al. [115], the impact of gamma radiation exposure on waste tire rubber was investigated with regard to its potential as a modified bituminous binder. Samples of crumb rubber were obtained from waste tires and irradiated with gamma rays at doses of 100, 200, and 300 kGy, at a dose rate of approximately 2.8 kGy/h. The irradiated CR samples were then combined with two different weight percentages of 5% and 10%. Results of FT-IR spectroscopy showed that the acrylonitrile butadiene macromolecules in CR samples lost the nitrile group, most likely due to an oxidation process by gamma rays, converting the cyanide group into O-H, C=O, and C-O groups. The study aimed to investigate the anti-aging performance and rheological properties of the modified crumb rubber binder. The anti-aging performance was evaluated using the Rolling Thin Film Oven Test (RTFOT), and it was discovered that increasing gamma radiation to 300 kGy decreased viscosity, making it easier for bitumen molecules to be absorbed (e.g., at 100 °C, the plastic viscosity decreased from 10,386 cp to 7393 cp for the sample with 10% 300 kGy-modified CR). The modified bitumen contains 10% CR irradiated at 300 kGy and exhibited a smaller decrease in penetration after RTFOT (92–93% residual ratio) compared to the non-irradiated binder (78–80% residual ratio), indicating less susceptibility to aging. The change in softening point after RTFOT aging decreases from 2.7 to 3.0 for CR-modified binders without irradiation and from 1.2 to 1.6 for the 300 kGy-irradiated modified binders, suggesting slightly better resistance to hardening upon aging. The irradiated CR at 300 kGy-modified binders retain higher ductility after RTFOT (83–89% of its initial value) compared to the non-irradiated CR-modified binders (62–68%), demonstrating better flexibility or lower temperature susceptibility after aging. Additionally, CR irradiated with 300 kGy-modified binder displayed significant improvements in stability, reducing the difference in softening point between the top and bottom parts of the sample in the phase separation test, from 4.0 °C to 0.2 °C, for the irradiated and non-irradiated CR-modified binders, respectively.

The study conducted by Hassan et al. [116] focused on materials composed of various compositions of maleic anhydride (MA)-treated reclaimed rubber powder (RRP) and natural rubber (NR), with the aim of exploring the potential of electron beam radiation cross-linking to enhance the properties of fiber-matrix composites. The study exposed the samples to irradiation doses of 30 and 50 kGy using an electron accelerator with an energy level of 1.5 MeV. The results showed that the tensile strength of non-irradiated composites increased gradually with the addition of glass fiber (GF), and the same trend was observed for irradiated composites at higher irradiation doses. The hardness of the composites also increased with increasing GF content, except at 50 kGy, where it decreased due to high degradation levels. Furthermore, the swelling percentage decreased with increasing RRP content, and the introduction of GF increased the hydrophobicity of the blends in motor and fluid oils. The swelling index increased with increasing GF content and decreasing irradiation dose, except for 50 kGy. The study suggested that the incorporation of GF into composites can significantly improve their thermal stability, and SEM images revealed good adhesion between the GF and polymeric matrix after exposure to electron beam irradiation.

Hassan et al. [117] also conducted a study on the radiative compatibilization of mechanochemically devulcanized waste rubber (DWR) from various tire types and its impact on the physical, mechanical, thermal, and morphological properties of virgin SBR using gamma irradiation. The devulcanization process involved introducing waste rubber particles between mill rolls with different devulcanizing chemical materials, including benzoic acid, zinc oxide, stearic acid, hydroquinone, and rosin. These materials were added to break the sulfur bonds in vulcanized rubber and render the sulfur passive. The blends of DWR were added to the SBR rubber, and the blends were exposed to gamma radiation using a ^60^Co gamma rays. The total integral doses used were 25, 50, 100, and 150 kGy, with a dose rate of 3.3 kGy/h. The gel fraction increased with a higher radiation dose. The best tensile value was obtained for a DWR ratio of 20% irradiated with 100–150 kGy. Elongation at break increased within the dose range 50–150 kGy for up to 20% DWR, and the elastic modulus of virgin SBR improved with radiation dose. The optimal abrasion resistance was observed for SBR loaded with 30% DWR. The solubility % decreased with increasing DWR content and showed the least value of brake oil resistance at DWR ratio of 30%.


**
*
Microwave radiation
*
**


Microwave radiation can destroy the vulcanization network on the surface of rubber particles, thereby enhancing the surface activity of the rubber. This process leads to an improvement in the viscoelasticity and storage stability of modified bitumen [43]. According to Hirayama’s research [44], microwave radiation resulted in the disintegration of polysulfidic bonds and the decomposition of chemical groups that had sulfur. However, the monosulfidic bonds with higher energy were not affected and remained intact (Figure 12). The study [45] also found that the intensity of the C–N functional group increased after microwave irradiation, which can prompt interactions and alleviate separation.

Hosseinnezhad et al. [118] conducted a study to investigate a sustainable hybrid method combining microwave radiation and bio-modification to modify crumb rubber for use in bitumen. The method promotes the grafting of polar bio-molecules (bio-oil from animal-derived waste) onto the rubber surface, creating surface-activated rubber, which aims to reduce segregation and workability issues. To create microwave-activated crumb rubber (M-CR), 60 g of crumb rubber was exposed to 400 W of microwave radiation for 4 min. The resulting surface-activated crumb rubber (SAR) exhibited improved properties, including a reduction in viscosity and segregation index compared to conventional crumb rubber-modified (CRM) bitumen. SAR binders also showed improved thermo-mechanical properties. The hybrid process offers a sustainable solution to mitigate segregation and workability issues while enhancing bitumen performance by altering rubber particle surface polarity and promoting secondary network formation. In turn, Lyu et al. [119] studied the potential of combining bio-derived molecules with CR particles to produce bio-modified rubberized bitumen with improved resistance to thermal and ultraviolet aging. Microwave radiation was used to graft waste cooking oil molecules onto rubber particles. The research found that waste cooking oil and rubber work together to enhance the aging resistance of rubberized bituminous binder. Bio-modification of crumb rubber particles improved their interaction with the binder, resulting in reduced aging. The bio-oil also helped maintain a higher healing capacity in the bitumen after aging. Although the bio-modified rubberized binder showed promising results against thermal and ultraviolet aging, further research is needed to study its performance under combined aging and the impact on moisture resistance in asphalt pavements. The combination of waste cooking oil and scrap tire crumb rubber, as additives to bitumen, leads to improved aging resistance, promoting sustainability in construction (Figure 13).


**
*
Chemical grafting
*
**


Xie et al. [120] improved the storage stability and performance of the modified binder by grafting acrylamide onto the surface of crumb rubber (CR) using solution polymerization. The process involved purifying CR with acetone and then activating it through a chemical grafting method using potassium persulfate as the initiator. An orthogonal test was designed to optimize the preparation process, and the study found that rubber content was the most significant processing parameter. The optimized preparation technology included a rubber content of 20%, a shear temperature of 170–190 °C, and a shear time of 90 min. The grafting activation of rubber improved the storage stability and compatibility of modified bitumen by enhancing the interaction of the mixture system through a chemical reaction between amide groups in rubber and acid groups in bitumen. The findings regarding BBR test–stiffness modulus *S* and creep rate *m* at various temperatures indicate that activated CR enhances the resilience and stress relaxation capacity of modified bitumen in colder conditions, thereby enhancing its resistance to deformation at low temperatures. The bitumen with CR activated by acrylamide showed a better storage stability than non-activated CR-modified bitumen. Difference in softening point ranged from 6.0 to 0.3 (for samples stored by 12 h) and 17.2 to 6.0 (72 h). The results from tests including viscosity at 175 °C, complex modulus, phase angle, and rutting factor indicate that activated crumb rubber enhances the modified bitumen’s capacity to withstand dynamic loads and recover from deformation, thereby enhancing its high-temperature rheological properties. Other studies [121] investigated the use of two methods to activate CR, coating by polyamide and grafting by acrylamide. The results showed that both activation methods improved the high- and low-temperature performance of the bituminous binder by enhancing the interaction between the rubber and binder; however, grafting activation by acrylamide was found to be more effective due to the chemical reaction between groups of acrylamide and binder.

Jiang et al. [122] proposed a modified method of dynamic vulcanization to prepare blends of ground tire rubber (GTR) and high-density polyethylene (HDPE) by combining surface devulcanization of GTR and in situ grafting technology. The process involved the mechanochemical devulcanization of GTR using intense shear and tetraethylenepentamine (TEPA), followed by grafting amine groups to the surface of devulcanized GTR (d-GTR) and adding grafting monomers of styrene (St) and glycidyl methacrylate (GMA) and initiator of DCP to the d-GTR/HDPE blends during the melt blending process. The study investigated the effects of initiator and grafting monomers on the mechanical, crystallization, and rheological behavior of the blends. Results showed that the modified method improved the compatibility and mechanical properties of the blends, which could be tailored by adjusting the initiator/grafting monomers ratio. The results of ATR-FTIR confirmed the devulcanization of GTR and the occurrence of grafting and cross-linking reactions between d-GTR and HDPE. The blends exhibited enhanced interfacial compatibility, and their crystallization behavior was affected by the grafting and cross-linking reaction. The blends had stable processing and reprocessing ability.

Kocevski et al. [123] developed a method to modify bituminous binder by grafting ground rubber tire particles with acrylic acid (AA). The grafting of acrylic acid on the GRT particles was shown to improve the bond between the GRT and bitumen, which can be speculated to be a result of increased surface area of the GRT particles and the formation of anhydride on the GRT surface. This resulted in improved binder properties, increased viscosity, and elevated failure temperature in some formulations. However, while the study concludes that there is a general increase, these effects can be influenced by a variety of factors, such as the degree of modification, the proportion of GRT added, the size of the GRT particles, and the type of base bitumen used. Additional research and testing would be necessary to fully understand the specific conditions under which these improvements occur.

### 5.4. Synthetic Rubbers

Ethylene propylene diene monomer (EPDM) and ethylene vinyl acetate (EVA) are the polymers used in bitumen modification [124]. It is more economically attractive to use recycled materials instead of virgin materials.

EPDM can withstand large deformations due to its stable structure caused by the cross-linking of the diene monomer, which makes the bond difficult to break [125]. Other properties to consider when determining the service life of EPDM are its high level of ultraviolet (UV) resistance, resistance to weather damage, ability to prevent fatigue damage, and water resistance. According to Ito [126] and Zaharescu et al. [127], the chain degradation by gamma radiation starts on propylene, and so far the dominant feature that influences oxidation of irradiated EPDM is the constitutive ethylene/propylene ratio. Ghoreishi et al. [128] indicate that the addition of small amounts of the EPDM (3%) and hybrid nanoparticles of CNT masterbatch (0.1%) and bentonite nanoclay (1.5%) can highly enhance the viscoelastic behavior of the bitumen at elevated temperatures. The addition of EPDM increases the average molecular weight of the bitumen matrix and forms a dominant polymer network, making the binder more resilient to deformations under load at high temperatures.

The semicrystalline EVA copolymer provides the modification of bitumen through the crystallization of rigid three-dimensional networks within the bitumen. Different crystalline structures are formed at different temperatures and have influence on conventional penetration, softening point, Fraass breaking point, ductility, and high temperature viscosity [129]. Waste EVA has good compatibility with bitumen, so it has been widely studied and applied. The results have shown that the large volume of the vinyl acetate group becomes a non-crystalline area or amorphous area, which plays a role similar to rubber when EVA is mixed with bitumen. The crystalline area of EVA has high stiffness, which acts as a reinforcing bar, and greatly improves the high-temperature stability, low-temperature cracking resistance, and viscosity of modified binder [130]. It also exhibits certain improvements in low-temperature performance when small amounts of waste EVA are added (2–4%) [131]. Particularly, addition of EVA at these concentrations resulted in an increase in the m-value of the modified bitumen at low temperatures by 20% at 4% additive. Adding EVA to bitumen generally increases the G*/sin(δ) value, which translates to improved rutting resistance. This effect decreases with temperature.

Gad et al. [132] explored how gamma radiation affects the characteristics of a composite made from waste low-density polyethylene (LDPE), ethylene vinyl acetate (EVA), and bitumen. In the study, blends of waste low-density polyethylene (LDPE), ethylene vinyl acetate (EVA), and bitumen were shaped into 2 cm-thick molds. For the irradiation process, samples were exposed to gamma radiation in air from a ^60^Co source, at a dose rate of 5 kGy/h. The total radiation dose applied ranged from 25 to 125 kGy. The best mechanical properties were reported at an integral irradiation dose of 75 kGy at a weight composition ratio of bitumen:EVA:waste LDPE blend, 1:1:1, where irradiation nearly doubles the tensile strength. A significant reduction in elongation at break by circa 60% and ca 20% increase of hardness suggests a higher resistance to deformation. Also, heat deformation is decreased by 50%. The decrease in elongation at break, by about 60%, potentially limits the material’s applicability in scenarios where flexibility and the capacity to absorb energy through deformation are critical. Nevertheless, the cited research demonstrated the feasibility of employing radiation compatibilization techniques on the polymer-bitumen blends. This approach has shown promise that with certain polymer blends, the option of pre-compatibilization by radiation followed by further high-temperature processing presents a viable pathway [133]. Ultimately, this strategy also allows a method for the application of gamma radiation or electron-beam technologies for enhancing pavements post-construction. Such techniques are already utilized on a smaller scale in industry, notably in the curing of automotive component lacquers [134]. This indicates the possibility of using high-energy radiation sources to reinforce pavements, contingent upon the success of these technologies in experimental settings.

Grigoryeva et al. [135] investigated the effects of gamma and electron beam irradiation on the properties of thermoplastic elastomers (TPEs) produced by dynamically vulcanizing blends of recycled high-density polyethylene (HDPE), ethylene/propylene/diene monomer (EPDM) rubber, and ground tire rubber (GTR) pre-treated with bitumen. Radiation treatment aimed to enhance the mechanical performance and radiation stability of these TPEs. Two specific formulations containing bitumen were tested: a mixture of HDPE/EPDM/GTR/bitumen with weight ratios of 40/35/12.5/12.5 wt.% and 40/35/17/8 wt.%, respectively. The mixtures were compared to the respective ones not containing bitumen. The TPEs were irradiated using a ^60^Co source with an energy of 1.25 MeV. The dose rate varied between 2.27 and 28 kGy/h, with a full absorbed dose range of 100 to 500 kGy. For electron beam treatment, an accelerator radiation was used with an energy of 3 MeV and a full absorbed dose of 100 kGy. The study found that irradiation significantly modifies tensile properties of the TPEs (tensile strength doubles and elongation at break drops 50%), suggesting potential for modification of the recycled polymer waste for practical applications. Bitumen serves a dual role in this context: it acts as a curing agent for the rubber components of TPEs (EPDM and pre-devulcanized GTR) and as an effective compatibilizer for blending the components together. This dual functionality is crucial for improving the mechanical properties of TPEs, as it facilitates better adhesion between the blend components, thereby enhancing the overall performance of the material. The use of bitumen pre-treated with GTR not only improves the tensile properties compared to bitumen-free TPE but also demonstrates stability to gamma irradiation, which is beneficial for applications requiring durability and longevity.

The research conducted by Kumar et al. [125] focused on assessing bitumens and asphalt mixes utilizing waste Ethylene–propylene–diene–monomer (EPDM) rubber as a modifier, sourced from various industries and automotive repair/service stations. Four EPDM content levels (2%, 4%, 6%, and 8% by binder weight) were tested for modification, with and without a cross-linking agent (sulfur) to enhance storage stability. The results of temperature sweep and frequency sweep tests showed that the presence of EPDM increased the complex modulus and decreased the phase angle, so it improved the stiffness and elasticity of the binders, leading to increased rutting resistance. This was also confirmed in MSCR tests and asphalt mixture wheel tracking tests. Specifically, EPDM dosages of 2%, 4%, 6%, and 8% brought reductions of 18%, 22%, 34%, and 38% in rut depth, respectively, compared to the control mix. In another approach, microwave irradiation was additionally applied [136]. The main goal of this study was to determine the phase separation (storage stability) of rubberized binders with non-tire rubber (EPDM rubber) particles with and without tire pyrolysis oil (TPO) and plastic pyrolytic oil (PPO) incorporation. EPDM rubber granules were subjected to microwave irradiation in a microwave oven at 700 watts and 2450 MHz for 2 min. Microwave pre-treatment of rubber granules with pyrolysis oils facilitated the development of composite-modified binders with excellent storage stability and rheological performance. Moreover, the percentage change in ΔIC=O and ΔIS=O between the initial day (day 0) and after storage for 14 days showed increased aging resistance of EPDM-modified binders containing TPO and PPO.

## 6. Discussion

From an asphalt pavements contractor’s perspective, maintaining the homogeneity of modified bitumen during storage is crucial, especially in response to abrupt meteorological shifts. Adverse weather conditions can often delay paving activities, necessitating the storage of modified bitumen in mixing plant tanks for extended periods, ranging from a few days to several. Ensuring homogeneity during these periods is vital to maintain the quality and workability of the asphalt mixtures. Conversely, from the viewpoint of road managers and users, enhancing the aging resistance of bituminous binders is a key factor. Improved aging resistance contributes to prolonging the service life of road pavement structures and minimizing the frequency and intensity of required maintenance and repairs. Moreover, the integration of recycled modifiers in bitumen production presents a significant opportunity for reducing the environmental impact of asphalt pavements.

As the popularity of additives for enhancing bitumen performance and recycling grows, radiation and radical grafting of polymers present promising techniques for enhancing the performance of bituminous binders. Along with the additives proposed, there is growing demand for compatibilizers. For effective compatibilization, especially in complex systems like bitumen, it is crucial to use modifiers that can interact with both aliphatic and aromatic phases or other specific phases present in the blend. Amphiphilic modifiers, to bridge between aliphatic and aromatic phases, are particularly valuable in this regard. In the context of bitumen modification, the goal is often to blend in polymers or additives that can integrate with the bitumen’s complex mixture of hydrocarbons. The challenge lies in achieving a stable dispersion of these additives, as demonstrated by the effective elimination of the naturally occurring heterogenic “bee-like” structures through chemical modification [91].

Many compatibilizers, like PE-g-MA, are produced via radical polymerization. This process allows for the introduction of absolutely non-compatible functional groups onto the polymer backbone, by covalent bonding, thereby combining the not combinable. Radiation, such as gamma or electron beam irradiation, is an innovative approach used to induce grafting and cross-linking in polymers. This method can be applied directly to the polymer blend, potentially simplifying the manufacturing process by reducing the need for additional chemical additives. However, these methods are still in the developmental stage and may have limitations in terms of scalability and control over the grafting process. Modifying the surface properties of polymers is another strategy to improve blend stability. This can be achieved through various chemical or physical treatments. The utilization of high-energy ionizing radiation in polymer processing offers an economical advantage, as the energy is directly channeled into radical reactions rather than being dissipated as heat. This efficient energy use not only facilitates the desired chemical modifications but also contributes to cost-effectiveness in the manufacturing process.

Emerging methods like radiation-induced grafting and the use of amphiphilic modifiers offer promising avenues for future research and development in the area of bitumen modification.

## Figures and Tables

**Figure 1 materials-17-01642-f001:**
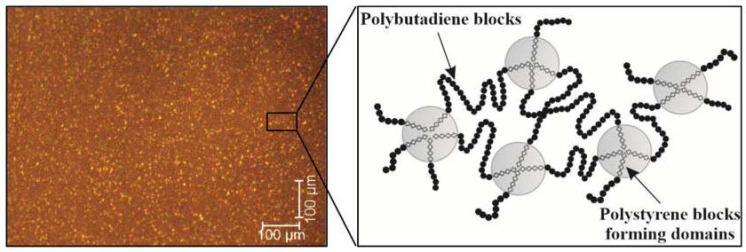
A microscopic image of morphology of SBS-modified bitumen (**left**) and a schematic representation of how the molecular structure of the SBS copolymer translates to the observed morphology (**right**).

**Figure 2 materials-17-01642-f002:**
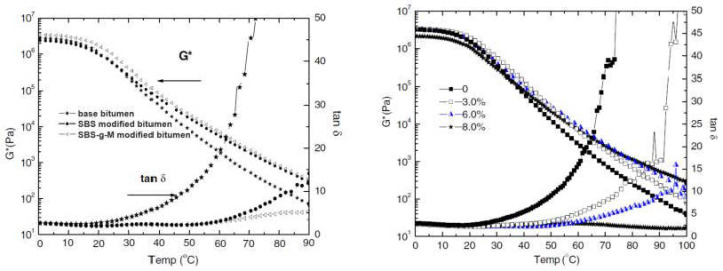
(**Left panel**): Complex modulus (G*) and tan δ versus temperature for base and modified bitumen at 10 rad/s. (**Right panel**): Effect of SBS-g-M content on dynamic mechanical properties of PMB. Adapted from [27] with the permission of Elsevier, copyright 2006.

**Figure 3 materials-17-01642-f003:**
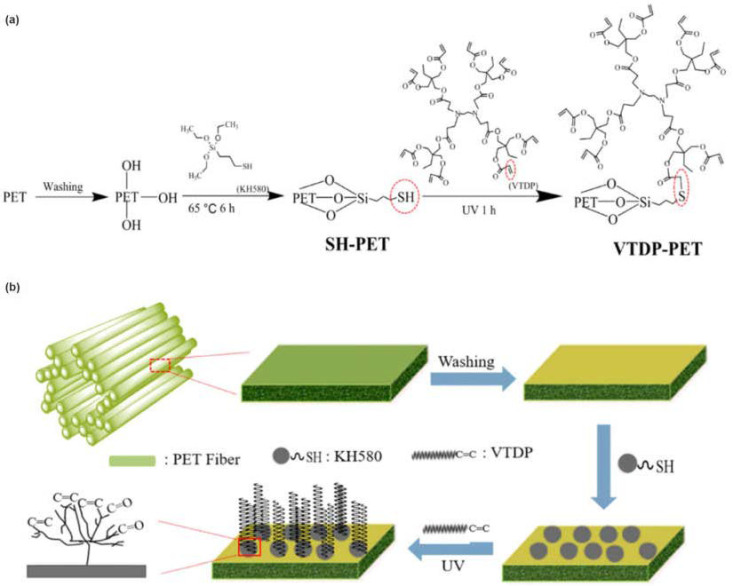
Reaction scheme (**a**) and flowchart (**b**) of vinyl-terminated dendritic polyester-polyester (VTDP-PET) fiber preparation. Reprinted from [61] with the permission of Wiley, copyright 2021.

**Figure 4 materials-17-01642-f004:**
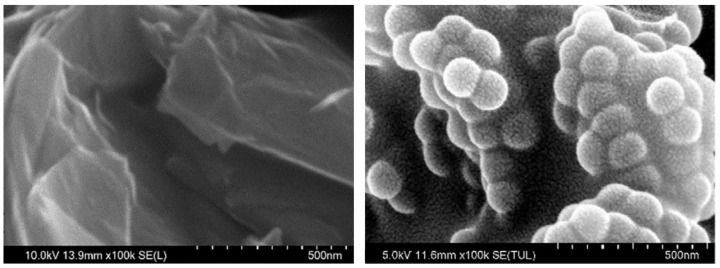
SEM of GNPs (**left**) and PS-GNPs (**right**). Adapted from [70] with the permission of Elsevier, copyright 2018.

**Figure 5 materials-17-01642-f005:**
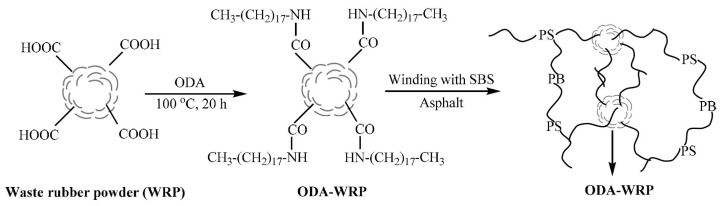
Formation mechanism of the ODA-WRP/SBS network structure. Reprinted from [71].

**Figure 6 materials-17-01642-f006:**
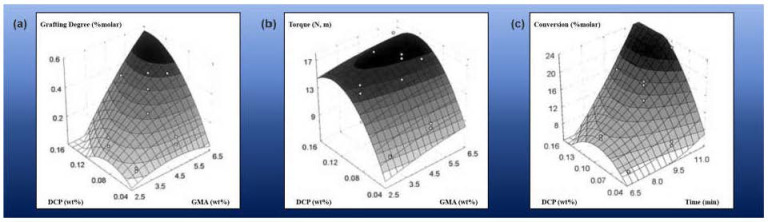
Effect of % dicumyl peroxide (DCP) and % glycidyl methacrylate (GMA) on grafting degree (**a**); effect of % DCP and % GMA on final torque (**b**); effect of % DCP and reaction time on conversion during the synthesis of GMA-grafted SBS (**c**). The white dots represent experimental data points. Adapted from [63], with the permission of Springer-Verlag, copyright 2001.

**Figure 7 materials-17-01642-f007:**
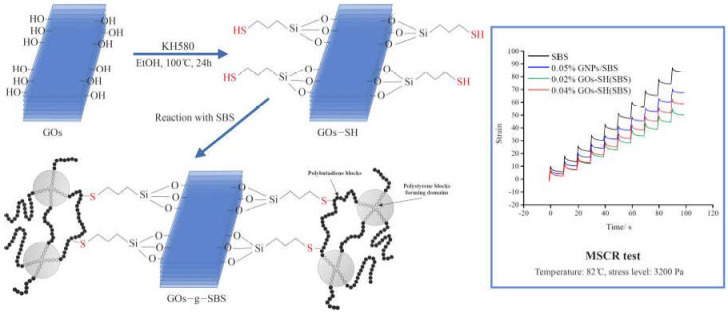
Preparation and properties of SBS-g-gOs-modified bitumen based on a thiol-ene click reaction in a bituminous environment. Adapted from [57].

**Figure 8 materials-17-01642-f008:**
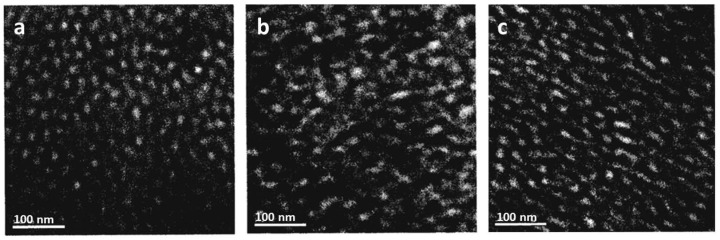
TEM images of SBS and end-functionalized SBS: (**a**) SBS, (**b**) SBS–COOH, and (**c**) SBS–N. Adapted from [73] with the permission of Wiley, copyright 2007.

**Figure 9 materials-17-01642-f009:**
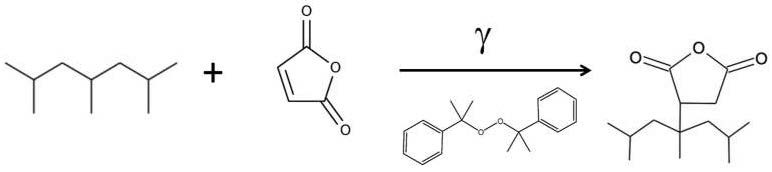
Grafting of waste polypropylene (PP_R_) with maleic anhydride (MA) with the use of dicumyl peroxide as an initiator and irradiation with gamma rays [42].

**Figure 10 materials-17-01642-f010:**
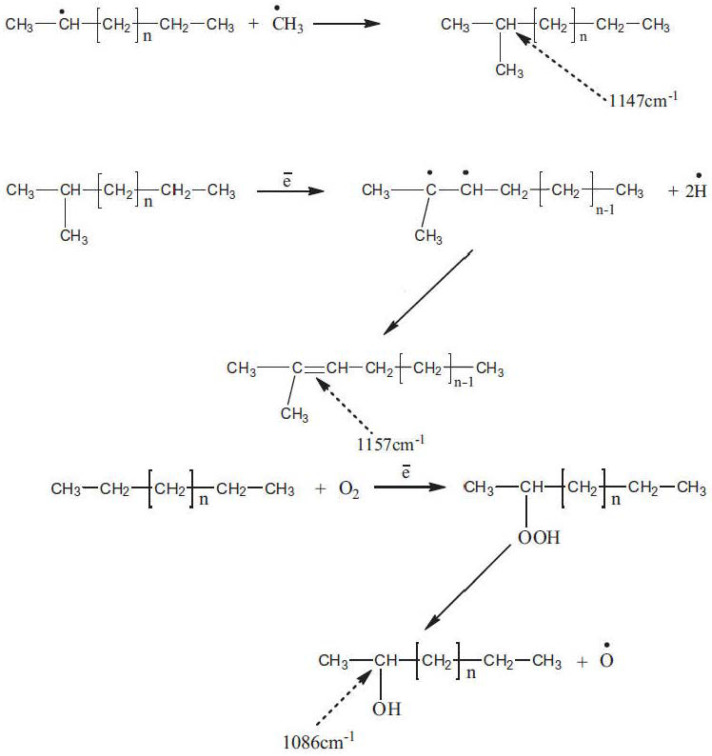
Scheme of chemical transformation in the recycled LDPE polymer chain under the influence of electron beam irradiation. First process illustrates the formation of tertiary carbon atoms during LDPE_R_ aging, before irradiation. Adapted from [41] with the permission of Elsevier, copyright 2014.

**Figure 11 materials-17-01642-f011:**
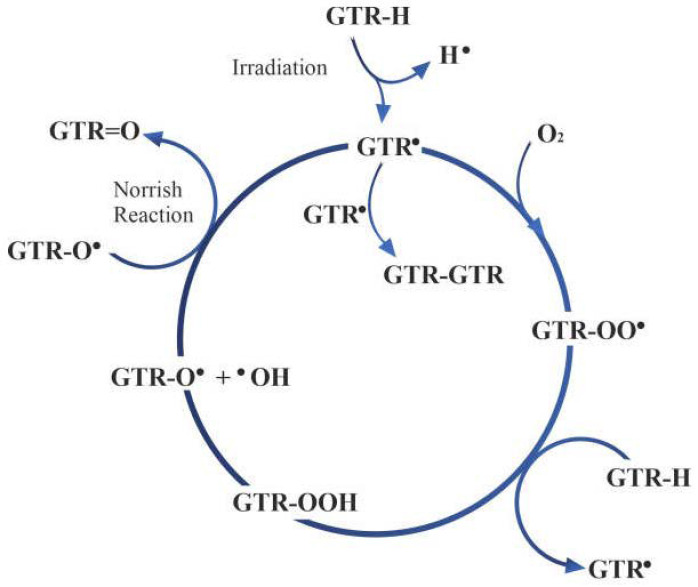
A depiction of the process by which radiation triggers oxidation in GTR particles [102].

**Figure 12 materials-17-01642-f012:**
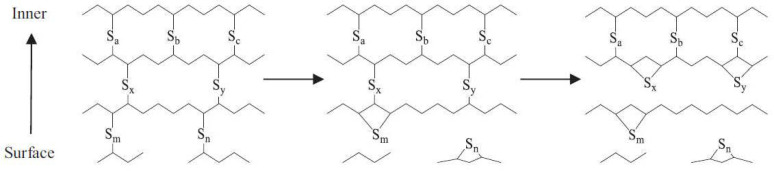
The disintegration of polysulfide bonds in CR following exposure to microwave radiation. Reprinted from [29] with the permission of Elsevier, copyright 2021.

**Figure 13 materials-17-01642-f013:**
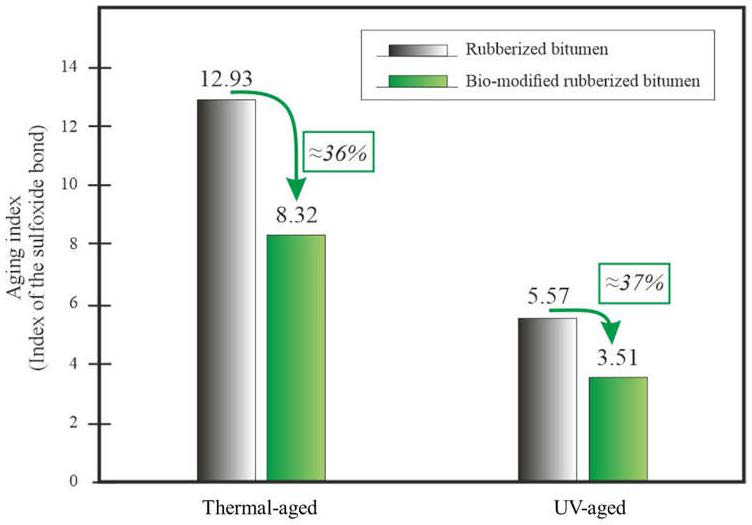
Bio-modification with waste bio-oil significantly enhances aging resistance of rubberized bituminous binder. Adapted from [119] with the permission of Elsevier, copyright 2021.

## Data Availability

No new data were created or analyzed in this study. Data sharing is not applicable to this article.

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
