# Peer review of "Radiation and Radical Grafting Compatibilization of Polymers for Improved Bituminous Binders—A Review"

_materials, 2024, doi:10.3390/ma17071642_

Round 1

Reviewer 1 Report

Comments and Suggestions for Authors

This article reviews the current research on new methods for improving the performance of asphalt binders through radiation and polymer modifiers. These methods mainly address issues such as high cost, low aging resistance, and storage stability, especially the key challenge of polymer/asphalt phase separation. These advanced modification technologies are expected to improve the performance and durability of adhesives, reduce the carbon footprint of transportation systems, and achieve sustainability.

Overall, this review provides a comprehensive understanding of how to improve the performance of asphalt binders through innovative and sustainable means.

The language in the paper is clear, accurate, and meets the standards of academic writing. Meanwhile, the keywords in the paper also appropriately reflect its theme and content.

Herein with several suggestions:

1. There is too much introduction to the current situation of asphalt, and the first three paragraphs should be simplified to quickly get to the topic.

 2. Line 247: This should be Figure 3 instead of Figure 2.

3. After chapter 2 can you introduce the indicators for evaluating asphalt bonding performance?

4. In Chapter 5 SBS modified asphalt and waste rubber can be improved through radiation and chemical methods, but there is no introduction to chemical methods for waste polypropylene and waste polyethylene. Is there currently no corresponding technology?

5. In chapter 4 and 5 Can images be added to visually demonstrate the improvement of asphalt bonding performance by two methods?

6. Add references from recent years to help readers understand the latest methods for enhancing asphalt bonding performance

Reviewer 2 Report

Comments and Suggestions for Authors

This paper give a review of 'Radiation and Radical Grafting Compatibilization of Polymers for Improved Bituminous Binder Performance'. It can be accepted after a minor revision.

(1) The title can be changed as 'A review of Radiation and Radical Grafting Polymers for Bituminous Binder'.

(2) The authors need to review the performance changes of Bituminous Binder after using Radiation and Radical Grafting Polymers.

(3) The use of images in this paper requires the consent of the publishers. 

Comments on the Quality of English Language

Some minor errors should be corrected. 

Reviewer 3 Report

Comments and Suggestions for Authors

Article "Radiation and Radical Grafting Compatibilization of Polymers for Improved Bituminous Binder Performance" is devoted to the performance of asphalt binders. The following questions and comments were found during the study of this article:

- Line 28 – if there is an indication of natural bitumen, it is necessary to explain what types there are. For example, article 10.3390/molecules28052065 provides a detailed scheme and description of the types of natural bitumen

- The use of polymers to modify bitumen binders undoubtedly plays a big role in increasing the service life of the road surface, as well as improving its mechanical characteristics. However, the introduction lacks part of the rationale for what modifiers other than polymer ones are used for this (for example, environmentally friendly renewable natural materials (10.1016/j.carbpol.2023.120896)), and why polymer additives are better. It would make sense to add this before the main part

- It would be good to separately decipher all the abbreviations - it will be easier for the reader to perceive the information

- Authors should spend more time editing the design of the article:

1) make the same line spacing between the heading and the main paragraph test, as well as between captions to pictures and the text after them (for example, as in the case of a Fig. 3)

2) Fig. 2 right picture - correct “Temp_(°C)”

3) Fig. 3 ainaccurate chemical formulas

4) Fig. 7 – a dot is missing before the word “Adapted”

5) Line 528 – “;” instead of "."

6) Line 570 – dot is missing

7) Line 710 – the figure number is bold

8) Fig. 13 is unreasonably large and unclear, it should be corrected

9) other numerous typos and inaccuracies

- There are many reviews devoted to the modification of bitumen using polymer additives (for example, 10.1016/j.eurpolymj.2014.02.005 and 10.3390/app9040742). Authors need to cite them and indicate how their review is different and of scientific interest (for example, it reviews more recent scientific literature, addresses unanswered questions, etc.).

Round 2

Reviewer 3 Report

Comments and Suggestions for Authors

The authors have made all the necessary edits to the article, so it can be published in its current form